# Advanced Open-Celled Structures from Low-Temperature Sintering of a Crystallization-Resistant Bioactive Glass

**DOI:** 10.3390/ma12223653

**Published:** 2019-11-06

**Authors:** Hamada Elsayed, Acacio Rincon Romero, Devis Bellucci, Valeria Cannillo, Enrico Bernardo

**Affiliations:** 1Department of Industrial Engineering, Università degli Studi di Padova, 35122 Padova, Italy; hamada.elsayed@unipd.it (H.E.);; 2Ceramics Department, National Research Centre, Cairo 12622, Egypt; 3Dipartimento di Ingegneria “E. Ferrari”, Università degli Studi di Modena e Reggio Emilia, Via Vignolese 905, 41125 Modena, Italy; devis.bellucci@unimore.it (D.B.); valeria@unimore.it (V.C.)

**Keywords:** bioglass, crystallization-resistant bioactive glasses, sintering, foams, additive manufacturing, scaffolds

## Abstract

Most materials for bone tissue engineering are in form of highly porous open-celled components (porosity >70%) developed by means of an adequate coupling of formulations and manufacturing technologies. This paper is dedicated to porous components from BGMS10 bioactive glass, originally designed to undergo viscous flow sintering without crystallization, which is generally known to degrade the bioactivity of 45S5 bioglass. The adopted manufacturing technologies were specifically conceived to avoid any contamination and give excellent control on the microstructures by simple operations. More precisely, ‘green’ components were obtained by digital light processing and direct foaming of glass powders suspended in a photosensitive organic binder or in an aqueous solution, activated with an organic base, respectively. Owing to characteristic quite large sintering window of BGMS10 glass, sintering at 750 °C caused the consolidation of the structures generated at room temperature, without any evidence of viscous collapse.

## 1. Introduction

Despite being useful in particulate form (e.g., embedded in polymer matrices) [1,2], bioactive glasses are mostly used in the form of scaffolds, with stochastic and non-stochastic open porosity (foams and reticulated scaffolds, respectively) [3]. The manufacturing of these structures is intrinsically conditioned by the control of viscous flow sintering of fine glass particles. More precisely, pressureless glass sintering occurs 50–100 °C above the dilatometric softening temperature [4] (corresponding to a viscosity of 10^11.5^ Pa·s), in turn placed slightly above the transition temperature T_g_ (but well below “Littleton” softening point, indicating gross viscous flow, for a viscosity of approximately 10^6.6^ Pa·s) [5] and often overlapping with the crystallization temperature [6]. The overlapping is also stimulated by the use of fine particles possessing a wide specific surface: in case of glasses sensitive to surface-induced crystallization, the crystallization temperature is so downshifted, with decreasing powder size, that the devitrification practically nullify the viscous flow [6] (rigid inclusions provide a significant increase of viscosity in softened glass [7]).

Sintering with concurrent devitrification (‘sinter-crystallization’), for 45S5 bioglass^®^, has a fundamental drawback in limiting the bioactivity [8]. This negative aspect may be overcome by revising the chemical formulation, i.e., producing new bioactive glasses, with two distinct aims: (i) adjusting the amount and nature of crystal phases developed upon firing, in order to achieve the bioactivity of 45S5 with a semi-crystalline material (e.g., Biosilicate^®^ glass-ceramic, developed by Zanotto and coworkers [9,10]); (ii) realizing a quite large temperature gap (‘processing window’) between softening and crystallization, in ‘crystallization-resistant bioactive glasses’ (including the well-known 13-93 bioactive glass) [11,12,13]. BGMS10 glass, the object of the present studies, belongs to the latter type [14,15].

Compared to 45S5 bioglass, the bioactive glass BGMS10 contains a lower content of alkali oxides, but it features the presence of magnesium and strontium oxides, thus bringing extra ions (Mg^2+^ and Sr^2+^) with a well established positive impact on biological properties [16]. Previous tests have shown that BGMS10 behaves very similarly to 45S5 in terms of in vitro bioactivity and in vivo biocompatibility [14,15,17].

Once a glass with low crystallization tendency and comfortable processing window is selected, attention must be paid to the manufacturing technique adopted for the obtainment of highly porous, open-celled components, matching the microstructural requirements of materials for bone tissue engineering (e.g., pores well-defined interconnections, with a diameter exceeding 100 μm, to allow effective tissue ingrowth and vascularization [18]).

A key condition is that the applied technology should not interfere with the microstructural evolution of glass powders upon thermal treatments; in other words, additives and/or particular processing conditions (e.g., use of very fine powders) should not determine changes in the crystallization tendency or cause a viscous collapse of the structures defined in the ‘green’ state.

The use of sacrificial organic binders is fundamental; among them, photocurable resins are undoubtedly precious in enabling the manufacturing of scaffolds with non-stochastic porosity by means of high-resolution stereolithography 3D printing [19]. When considering suspensions of fine glass particles in photocurable resin (i.e., a bi-phasic mixture), instead of photocurable liquid (a monophasic system), the resolution is obviously conditioned by the particle size and by light scattering at the liquid–particle interfaces [20,21,22]; anyway, highly uniform samples, even with complex geometries, were successfully obtained starting from glass undergoing sinter-crystallization [23,24]. The sudden increase of viscosity, caused by crystal precipitation, prevented any collapse of cellular structure upon firing. The shape retention had been observed also in stereolithography experiences with bioglass 45S5 [25,26]; however, in this specific case, samples exhibited quite poor mechanical properties and were partially crystallized (with a negative impact on bioactivity, as discussed above). The present investigation aims at demonstrating the suitability of stereolithography for the manufacturing of reticulated scaffolds with good shape retention upon firing, despite the absence of crystallization, owing to the use of BGMS10 bioactive glass. 

The use of fugitive organic binders is useful also for the obtainment of open-celled foams by gel-casting, i.e., by intensive mechanical stirring and incorporation of air bubbles (with the support of surfactants) in liquids undergoing progressive gelation. Widely applied to bioactive glasses from sol–gel approach [27,28] (operating on solutions of metallorganic compounds), the process has been successfully applied to suspensions of glass powders, subjected to gelation according to the addition of a quite complex mixture of organic agents, such as monomers, cross-linkers, and catalysts [29,30]. Air bubbles are stabilized by the characteristic gelation-induced pseudoplasticity, i.e., by the sudden increase of viscosity upon reduction of strain rate (low viscosity upon intensive stirring, high viscosity when stirring stops). Porous samples are formed already in the ‘green’ state (dried suspension) before final consolidation by viscous flow sintering.

The present investigation, besides discussing the application of stereolithography 3D printing to BGMS10, aims at disclosing the potential of a simplified approach to gel-casting of bioactive glass foams, deriving from recent experiences on the alkali activation of glass aqueous suspensions [31]. According to alkaline attack, gelation is achieved by formation of an inorganic binding phase (instead of an organic, polymeric phase), consisting of hydrated silicate and/or carbonate compounds, involving ions extracted from the glass (Ca^2+^) and/or from the alkaline activator (Na^+^, K^+^, from NaOH and KOH, respectively). Although highly effective (and flexible, i.e., open to a vast range of glasses [32,33,34]), the process has a substantial drawback in the contamination of the adopted glass operated by oxides in the binding phase, decomposed upon firing; in particular, a surplus of alkali oxides may modify both biological behavior and crystallization (it was observed that alkali activation may widen and shift the crystallization exothermic peak at lower temperature [32]). With the present paper, we will show the feasibility of a ‘hybrid’ activation, which recovers the concept of gelation as an effect of glass inclusion in a basic environment, but avoids contamination, owing to the use of an ammonium salt.

## 2. Materials and Methods

The chemical composition of BGMS10 glass is reported in Table 1. The glass batch was prepared from analytical grade powder reagents (from Carlo Erba Reagenti, Milano, Italy) and melted at 1450 °C in a Pt crucible, in air. The molten material was suddenly cooled by pouring into deionized water. The drastic quenching provided a number of fragments, subsequently left to dry at 110 °C for 2 h. The frit was later reduced into fine powders, with a size below 38 μm, by dry ball milling and manual sieving.

Samples with non-stochastic porosity were manufactured by using DLP (digital light processing), in analogy with previous experiences [23,24]. Fine glass powders were suspended in a commercially available photocurable acrylic polymer (Tripropylene glycol diacrylate, Robot Factory S.R.L., Mirano, Italy), already comprising a suitable photo-initiator and photo-absorber, for a solid load of 60 wt. %. The suspension was printed using a printer (3DLPrinter-HD 2.0, Robotfactory S.R.L., Mirano, Italy) operating in the visible light range (between 400 and 500 nm). After cleaning in an ultrasonic bath with isopropanol for 3 min, the samples were subjected to a secondary curing step, in a UV curing chamber (operating wavelength 365 nm, Robot Factory S.R.L., Mirano, Italy), for 15 min. Finally, the samples underwent a preliminary heat treatment in the air (1 °C/min heating rate) at 550 °C for 5 h, aimed at the burn-out of the photo-cured binder.

‘Green’ glass foam samples were prepared according to frothing of an aqueous suspension of fine glass powders, for a solid loading of 65 wt. %, activated by addition of tetramethyl-ammonium hydroxide (TMAH, (CH_3_)_4_NOH, reagent grade, Sigma-Aldrich, Gillingham, UK) in a concentration of 1 mol/L. Glass slurries were firstly kept under low-speed mechanical stirring (500 rpm) for 3 h to induce partial dissolution and gelation. Secondly, the slurries were foamed by vigorous mechanical mixing (2000 rpm) for 5 min after the addition of 4 wt. % Triton X-100 (polyoxyethylene octyl phenyl ether, Sigma-Aldrich, Gillingham, UK) and subsequently cast in polystyrene moulds (50 mm diameter). Finally, the foams were dried at 40 °C for 24 h, demoulded and subjected to a preliminary heat treatment in air at 350 °C (5 °C/min heating rate) for 1 h for the burn-out of organics.

Both reticulated and foamed scaffolds, after relative preliminary heat treatments, underwent a sintering treatment in air at 750 °C (with heating rate of 1 °C/min up to 500 °C, then with 5 °C/min up to 750 °C) for 1 h, followed by natural cooling.

The evolution of samples was investigated by means of Fourier transform IR spectroscopy (FTIR, FTIR model 2000, Perkin Elmer, Waltham, MA, USA), X-ray diffraction on powders (XRD; Bruker AXS D8 Advance, Bruker, Germany) and thermal analysis (thermogravimetry and differential scanning calorimetry, TG/DSC; Mettler Toledo TGA/DSC 3^+^, STARe System, Columbus, OH, USA).

The bulk density was computed from the weight-to-volume ratios on regular blocks (cut from bigger foamed samples or directly available from DLP), after careful determinations of weights and dimensions by means of an analytical balance and of a digital caliper, respectively. The apparent and true densities of the various samples were measured by He gas pycnometry (Micromeritics AccuPyc 1330, Norcross, GA, USA), applied on samples in bulk and powder forms. Morphological and microstructural characterizations were performed by optical stereomicroscopy (AxioCam ERc 5s Microscope Camera, Carl Zeiss Microscopy, Thornwood, New York, NY, USA) and scanning electron microscopy (FEI Quanta 200 ESEM, Eindhoven, The Netherlands). 

The compressive strength of the porous glass scaffolds was measured at room temperature by means of an Instron 1121 UTM (Instron, Danvers, MA, USA) operating with a cross-head speed of 0.5 mm/min. Each data point represents the average value of at least 10 individual tests.

## 3. Results

A first series of investigations concerned the application of stereolithography to BGMS10, starting from a diamond cell design recently tested for wollastonite-diopside glass-ceramics [24], as illustrated by Figure 1. The particular design, maximizing the bending of thin struts, was chosen specifically to verify the resistance of BGMS10 to viscous collapse upon firing.

The limited morphological changes from samples in the printed state, shown in Figure 1a,b, to samples after firing, illustrated by Figure 1c, clearly show that by operating at the lower limit for pressureless sintering (with T_g_ ~670 °C [14], the firing temperature —750 °C, i.e. T_g_ + 80 °C, was in the range of dilatometric softening temperature), a good shape retention could be achieved despite the absence of crystallization (in previous experiments with wollastonite-diopside glass-ceramics [24] the shape was maintained as an effect of viscosity enhancement operated by intensive crystal precipitation). BGMS10 was evidently a ‘long’ glass, i.e., not exhibiting an abrupt decrease of viscosity with increasing temperature above T_g_. The fully amorphous state is demonstrated in Figure 2, and it is consistent with the DSC curve of Figure 3, displaying the crystallization temperature far above the adopted sintering temperature.

All printed samples were opaque white as an effect of the presence of tiny micropores (see the small black dots in Figure 1d) and some roughness (Figure 1d still displays the former glass granules). The circles in Figure 1c actually evidence some defects likely originating from not optimized debinding, i.e., gas release, from the burn-out of the acrylic resin, overlapping with the sintering phase and leading to some cracks. Since BGMS10 did not undergo crystallization, the viscous flow provided some healing of the same cracks, saving the integrity of samples.

The saving of structural integrity can be understood from the results of mechanical testing. The diamond cell structure had been selected also as an example of ‘bending dominated’ design [24]. According to Gibson and Ashby [35], the crushing of brittle solids with such morphology, causing the systematic buckling of the struts, is described by a simple equation: σ_c_ = 0.2·σ_bend_·(ρ_rel_)^1.5^(1)
where σ_c_ represents the compressive strength, σ_bend_ is the bending strength of the solid phase, and ρ_rel_ is the relative density (ratio between geometric and true densities). The observed crushing strength (σ_c_ = 0.49 ± 0.03 MPa, see Table 2) cannot be considered as low; in fact, according to the low relative density (slightly exceeding 0.1, with a total porosity of nearly 90 vol. %), the bending strength of the solid phase, predicted by reversing Equation (1) (σ_bend_ = 5·σ_c_·(ρ_rel_)^−1.5^), is about 71 MPa, in excellent agreement with the bending strength values of silicate glasses [36] despite the presence of the above mentioned debinding defects.

Much stronger scaffolds (with a compressive strength well exceeding 7 MPa, see Table 2) were developed by using a second design, as illustrated by Figure 4. Cubic samples were derived by the overlapping of smaller cubic cells in a variable number (4 or 6 cells per side, as shown by Figure 4a and Figure 4b, respectively). In this case, the application of the model from Gibson and Ashby would not be appropriate, considering the much thicker struts and the lower porosity (from 50 to 62 vol. %). The debinding defects favored by the increased strut thickness, with the new design, were again healed by the viscous flow of BGMS10, as shown by Figure 4c.

The adopted designs, ‘thin-walled’ (diamond cell) and ‘thick-walled’ (cubic cell), may be seen as two ‘extreme’ solutions in DLP of BGMS10, useful to appreciate the successful combination of densification with retention of both shape and amorphous state. Additional efforts will be undoubtedly dedicated, in the near future, in the manufacturing of scaffolds with different strut length/thickness ratio, different overall geometry and adjusted debinding phase.

The second series of investigations concerned the effectiveness of activation of aqueous slurries by means of the adopted organic base, compared to alkali activation. Suspensions of glass powders in NaOH or KOH solution typically undergo gelation by formation of hydrated silicate and carbonate compounds [31,32,33,34] that could be detected by means of infrared spectroscopy and X-ray diffraction analysis.

Figure 5a shows a ‘green’ foam after the drying step. The new activation was successful in causing the gelation of the glass slurry and stabilization of air bubbles, incorporated by intensive mechanical stirring. According to Figure 5b, particles were effectively ‘glued’ together, in analogy with what found with NaOH or KOH solutions. However, the nature of the surface compounds did not correspond with that previously achieved.

The FTIR spectra in Figure 6 shows many bands in samples after activation (‘green’ foams) but absent either in the starting glass or in fired foams. Like in the case of NaOH and KOH activation [32], the bands at 3000–3700 and 1650 cm^−1^ are attributable to hydration of BGMS10 glass (O-H stretching and bending), but there is no clear carbonation (the C-O bond is usually detected by a well-defined peak centered at about 1450 cm^−1^) [31,32]. The surfactant is clearly distinguishable from the bands labeled with ‘S’ (particularly that 2900 cm^−1^). Contribution to the IR absorption came also from TMAH (see symbols in Figure 5, denoting major absorption bands [37]). The band at 1250 cm^−1^ may be attributed again to the activator, being reported for the C-N stretching vibration [38].

The numerous bands below 1300 cm^−1^ are quite interesting in revealing a more intense silica dissolution than in the cases of attack by NaOH or KOH (at low molarity). The band at 1120 cm^−1^ can be attributed to the stretching vibration of Si-O bonds in free SiO_4_ tetrahedra, as found in colloidal silica [39]; the bands centered at 900 cm^−1^ finds analogies with silica hydrated gels [40,41,42]. Finally, the sharp band standing at 970 cm^−1^ identifies the presence of Si-OH groups of the hydrated gel [43,44]. 

Unlike in previous cases, the activation with TMAH could not be recognized by formation of crystalline phases or by shifts in the peak position of the amorphous halo (typical of the X-ray diffraction of glasses), as shown by Figure 2. Furthermore, the firing at 750 °C again did not cause any crystallization. 

The thermogravimetric plot, shown in Figure 7a, revealed a continuous weight loss of the dried sample up to the sintering temperature, up to 15 wt. %, i.e., far exceeding the contribution of the surfactant (4 wt. %). Interestingly, the overall loss matched almost perfectly that of a bioactive glass (undergoing sinter-crystallization upon firing) previously activated with NaOH-containing solution (1 M) [32], reported in Figure 7b (also displaying the weight loss of Triton X-100). A substantial difference is represented by the fact that with TMAH-induced activation most of the weight loss occurred below 500 °C, while with NaOH more than one-fifth of the total loss was achieved above this threshold. ‘High-temperature losses’ with NaOH activation were ascribed to the decomposition of calcium silicate hydrated (C-S-H) compounds, known to release water vapor above 500 °C [45]. The losses for TMAH-activated BGMS10 are reasonably due to the evolution of physically adsorbed water (released below 200 °C) as well as condensation of silanol groups formed at the surface of glass particles, upon basic attack. We cannot exclude a minor contribution of organic moieties evolving from TMAH.

Previous experiments on glass–ceramic foams, from the activation of CEL2 bioactive glass [32], had revealed a substantial stimulation of crystallization, with the distinctive exothermic peak appearing broader and at much lowered temperature, interpreted as the effect of an alkali-enriched transient liquid layer from the decomposition of the binding gel on the surface of softened glass upon firing. The volumetric changes and ionic displacements in softened glass, associated with crystallization, could be somewhat catalyzed by a low viscosity extra phase (alkali rich glasses feature lower activation energy for crystal growth [46]), progressively dissolving in the same base glass. The differential scanning calorimetry plot shown in Figure 3 testifies that the crystallization tendency of BGMS10 was not significantly altered by the activation: the exothermic peak had a slight downshift but no broadening. A large exothermic effect was detected between 200 and 400 °C, consistent with the burn-out of organic moieties.

The firing of green foams confirmed the open-celled structure available in the green state (Figure 5). As shown by Figure 8 (left), wide openings between adjacent cells (above 100 µm) were kept, while the cell walls had intensive densification, by viscous flow of BGMS10, to become nearly transparent. Such transparency was also a confirmation of the complete burn-out of organic additives, much before sintering could take place. Higher magnification details (shown in Figure 8, right) confirmed the formation of thin (but hollow) membranes. The presence of these membranes and the absence of any crack motivate the far higher strength-to-density ratio of foamed samples, compared to diamond cell scaffolds.

The substantial lacking of any interference of the adopted shaping technologies on the evolution of BGMS10 bioactive glass suggests no change in the biological response compared to previous findings [14,15]. Anyway, cell tests will be considered, in the near future, for further validation. New investigations will include also the effect of secondary phases, which may benefit, in their stability, from the low sintering temperature. In particular, further studies will concern the use of BGMS10 mixed with hydroxylapatite powders, already tested for the manufacturing of dense discs [15]. 

## 4. Conclusions

The present study confirmed that BGMS10 bioactive glass, due to its characteristic processing window, could be applied for the manufacturing of highly porous open-celled components without viscous collapse. The viscous flow of BGMS10, although controlled, was effective in providing some healing of cracks developed upon firing of scaffolds with non-stochastic porosity due to the debinding of a commercial photocurable binder. The adopted digital light processing technique allowed for the easy obtainment of samples with different designs (to be further explored in the future). As an alternative to the printed scaffolds, open-celled foams (i.e., structures with stochastic porosity) could be obtained by a facile gel casting method, based on the activation of an aqueous suspension of glass powders, with a strong organic base. The gelation, unlike in previous experiments concerning the activation by means of inorganic bases (alkali hydroxides), had no impact on both chemistry and sintering of BGMS10.

## Figures and Tables

**Figure 1 materials-12-03653-f001:**
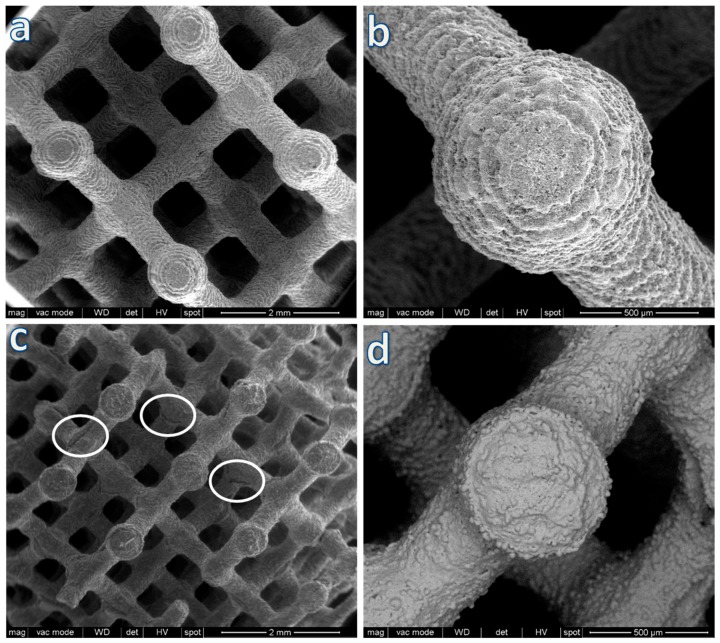
Microstructural details of a scaffold with diamond cell design after printing; (**a**,**b**) before firing and (**c**,**d**) after firing at 750 °C.

**Figure 2 materials-12-03653-f002:**
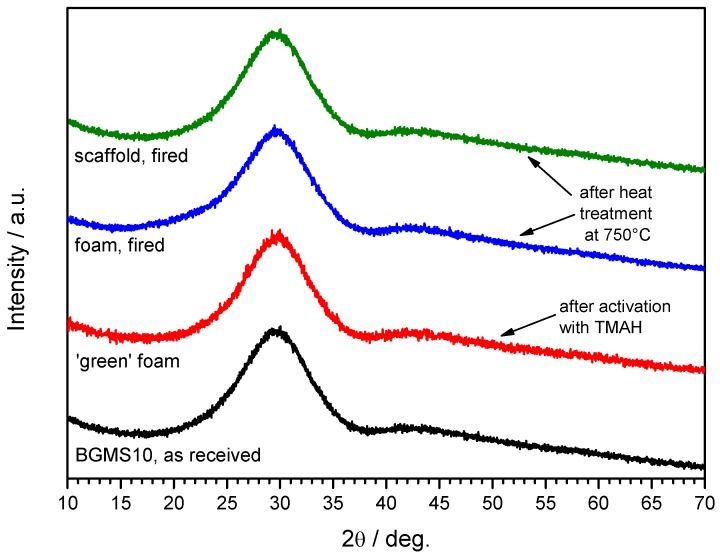
Mineralogical analysis of BGMS10 glass as received, activated with tetramethyl-ammonium hydroxide (TMAH) and after firing at 750 °C (foam and scaffold).

**Figure 3 materials-12-03653-f003:**
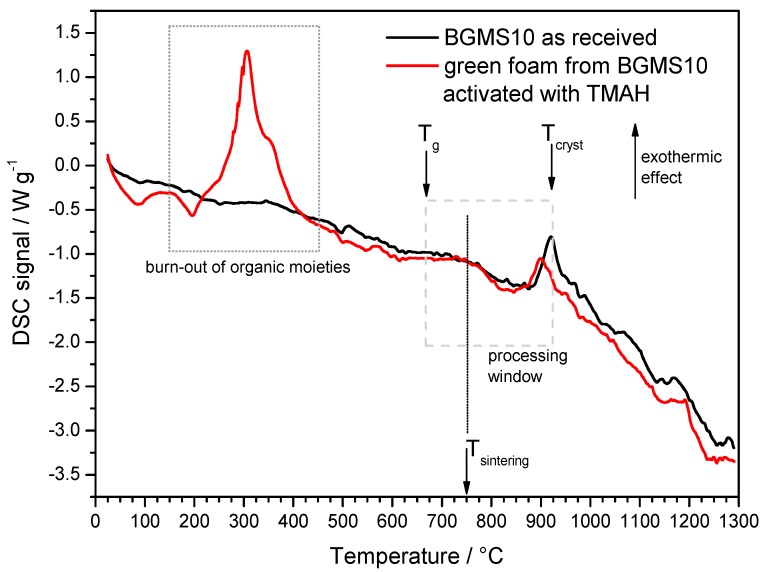
DSC plot of BGMS10 glass in the as received conditions and in the form of ‘green’ foam.

**Figure 4 materials-12-03653-f004:**
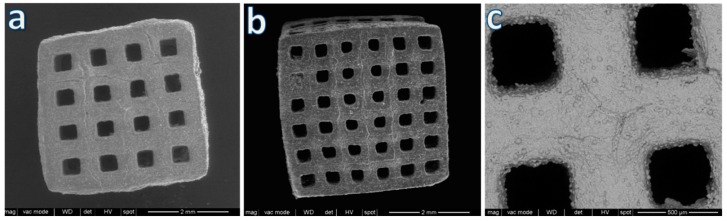
Microstructural details of samples from cubic cell design: (**a**) 4 cells per side; (**b**) 6 cells per side; (**c**) crack healing by viscous flow.

**Figure 5 materials-12-03653-f005:**
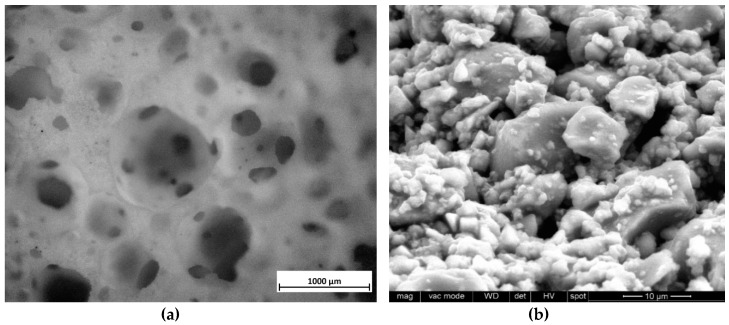
Green foam after activation of BGMS10 with TMAH: cellular structure (**a**, optical stereomicroscopy image); detail of cell wall (**b**, SEM image).

**Figure 6 materials-12-03653-f006:**
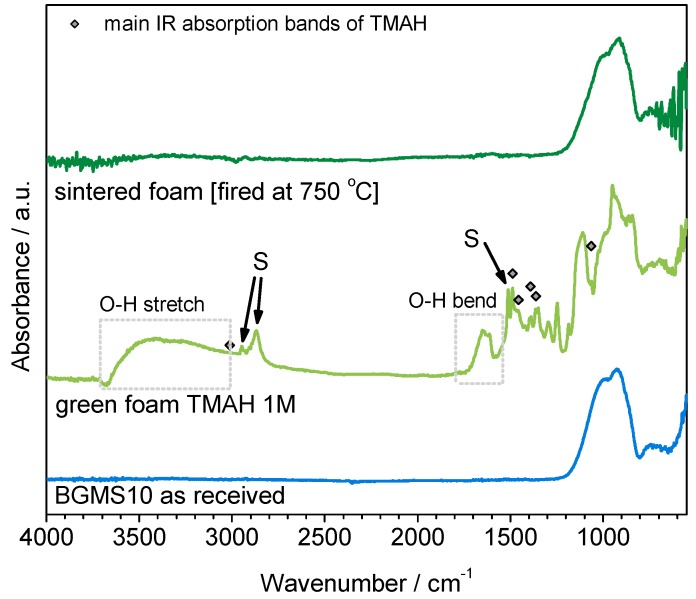
FTIR spectra of BGMS10 before and after activation and after firing.

**Figure 7 materials-12-03653-f007:**
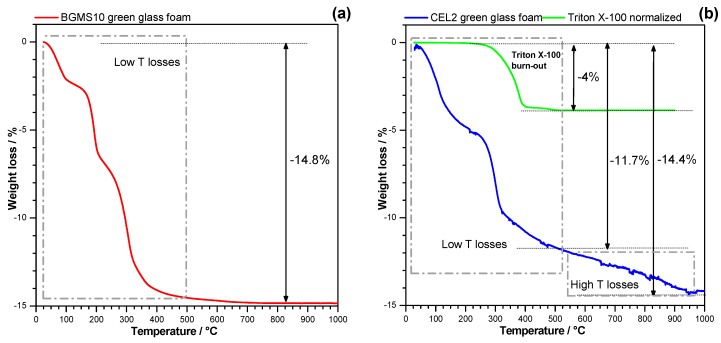
Comparison between BGMS10 glass activated with TMAH (**a**) and CEL2 glass activated with NaOH (**b**, adapted from Elsayed et al. [30]).

**Figure 8 materials-12-03653-f008:**
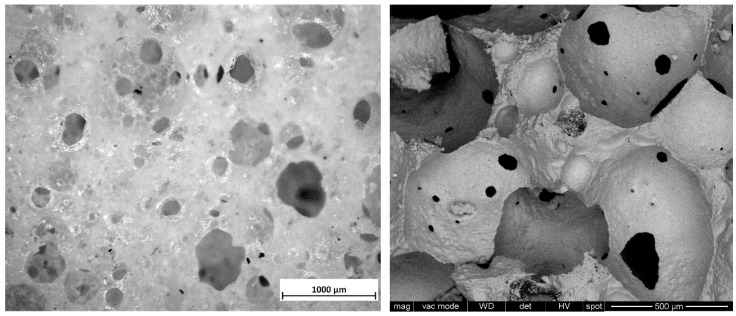
Fired foam: cellular structure (left, optical stereomicroscopy image); high magnification detail (right, SEM image).

**Table 1 materials-12-03653-t001:** Chemical formulation of BGMS10 bioactive glass.

Oxide	BGMS10 Glass	
	mol%	wt. %
SiO_2_	47.2	44.6
CaO	25.6	22.6
MgO	10	6.3
SrO	10	16.3
Na_2_O	2.3	2.2
K_2_O	2.3	3.4
P_2_O_5_	2.6	4.5

**Table 2 materials-12-03653-t002:** Summary of density and compressive strength data for all developed cellular glasses.

Cellular Structure		Bulk Density (g/cm^3^)	Apparent Density (g/cm^3^)	True Density (g/cm^3^)	Total Porosity (%)	Open Porosity (%)	Compressive Strength (MPa)
Reticulated scaffold	*Diamond*	0.31 ± 0.05	2.93 ± 0.02	3.00 ± 0.02	90	89.4	0.49 ± 0.03
	*Cubic, 4 × 4*	1.12 ± 0.04	2.93 ± 0.02	3.02 ± 0.02	63	61.8	7.67 ± 0.61
	*Cubic, 6 × 6*	1.39 ± 0.03	2.93 ± 0.02	2.96 ± 0.02	53	50.6	15.53 ± 1.30
Foam		0.57 ± 0.03	2.94 ± 0.02	3.17 ± 0.02	82	80.5	1.92 ± 0.10

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
