# Peer review of "Advanced Open-Celled Structures from Low-Temperature Sintering of a Crystallization-Resistant Bioactive Glass"

_materials, 2019, doi:10.3390/ma12223653_

Round 1

Reviewer 1 Report

This paper assesses the suitability of a previously report bioactive glass (by the same group) for 3D printing. Different designs are demonstrated, the glasses are sintered without crystallisation and their mechanical properties are assessed.

The authors should refer to their glass as a bioactive glass BGMS10 not as bioglass. The title must be changed as must all references to GMS10 bioglass. Bioglass® is a registered trade mark and all the other glasses need to be referred to as a bioactive glass not a bioglass. Similarly when reporting the literature they refer to the 13-93 bioglass, again this has never been called bioglass and is always referred to as 13-93 bioactive glass.

The novelty appears overstated and should be toned down, also although they briefly mention other glasses with improved thermal processing windows that do not crystallise (e.g. 13-93) however this need to be emphasised more because these glasses have been around a long time and have very similar compositions to the ones being presented here.

The introduction should make it much clearly that they have already presented papers on MBMS10 and should state how its bioactivity compares to 45S5, 13-93 etc and use this as a rationale for proceeding with their experiments.

TMAH needs to be described in full the first time it is mentioned (Tetramethylammonium hydroxide).

Line 127 with 5 1 degC (also there is an extra space before the degree symbol -

Line 141 is says the porous strength of the ceramics, but XRD shows it is still an amorphous glass not a ceramic.

Figure 1 caption, expand to say what parts a – d are in the figure caption

Table 2, are both the apparent and true density needed, there is no difference in these data sets.

Line 232, it says there is no evidence of a C-O bond, please add which would be expected to occur at .. cm-1 so the reader can be convinced.

272, replace remarkable with large peak

276the cell walls had a so intensive (remove the word so)

Sometimes there is a space between the number and the degree, sometimes there is not – be consistent,

Author Response

Dear Editors, dear Reviewers,

many thanks for the observations on the first version of the paper.

Please refer to the attached file, which summarizes our replies.

We are also submitting a revised version of the paper with changes hihglighted in red colour.

Many thanks for your attention

(on behalf of all coauthors)

Best regards

Enrico Bernardo

Reviewer 2 Report

This manuscript aims to describe the fabrication of bioactive bioglass with highly porous structures and good shape fidelity. Overall, the text is well written and scientifically robust.

However, the characterisation the 3D scaffolds is somewhat limited. Some of the major concerns are that the authors clearly state in the introduction that one main limitation is that bioglass often loses its bioactivity through these processing steps, which is not evaluated in this study. A higher impact would be achieved with some basic in vitro biology characterisation. In addition, a more detailed characterisation of the 3D structure would be beneficial, e.g. pore size before and after, prelevance of cracks, incidents of collapsed structures etc. As seen in figure 1, it appears that not only are cracks forming, the fibre has also collapsed fully in some areas. It should be noted if this is an artifacts from image analysis or from the fabrication process. The authors further mention that the organic material is burned off, but this is not clearly confirm though any of the analysis. How much residue is left and will it impact any downstream applications? Again, biological evaluation would be beneficial. Another major concern is that statistical analysation seems to be missing at it is unclear how many samples were fabricated and analysed. The reproducibility of the two processes should be evaluated and discussed more clearly. In addition to these major comments, a few minor edits should be considered:

Add details about the photo-initiator and photo-absorber used for the DLP step Add details about intensity and exact wavelength (nm) of the UV 110 chamber used The Figure 1 text is not referring to the donated A, B, C and D images. More details need to be added to understand which image is what. The samples in figure 1A and 1C appears to be fabricated with different pore sizes but this is not explained or discussed. Writing the conclusions as bullet points is not professional

Author Response

(The authors gave the same response as above.)

Reviewer 3 Report

Figure 1 caption would benefit from descriptions of a - d micrographs.

Author Response

(The authors gave the same response as above.)

Round 2

Reviewer 1 Report

the reviewers comments have mostly been addressed.

it would have been nice to have seen the in vitro characterisation as suggested by the other review